Composable languages for bioinformatics: the NYoSh experiment

Simi Manuele
Campagne Fabien fac2003@campagnelab.org
The HRH Prince Alwaleed Bin Talal Bin Abdulaziz Alsaud Institute for Computational Biomedicine, The Weill Cornell Medical College , New York, NY , United States of America
Department of Physiology and Biophysics, The Weill Cornell Medical College , New York, NY , United States of America
Crandall Keith
Electronic publication date: 2014 Jan 2
Publication date: 2014
Volume: 2
Electronic Location ID: e241
Received 2013 Nov 26; Accepted 2013 Dec 17
Copyright: © 2014 Simi et al.
Copyright year: 2014
Copyright holder: Simi et al.
License: This is an open access article distributed under the terms of the Creative Commons Attribution License, which permits unrestricted use, distribution, and reproduction in any medium, provided the original author and source are credited.
License URL: https://creativecommons.org/licenses/by/3.0/

Keywords: Language workbench, Language composition, NYoSh, GobyWeb, Scripting language

Funding: National Institutes of Health/National Institute of Allergy and Infectious Diseases R01AI107762-01 National Institutes of Health/National Center for Research Resources UL1 RR024996 This investigation was supported by grant UL1 RR024996 (National Institutes of Health (NIH)/National Center for Research Resources) of the Clinical and Translation Science Center at Weill Cornell Medical College and by grant R01AI107762-01 (National Institutes of Health (NIH)/National Institute for allergies and infectious diseases, to Campagne F). The funders had no role in study design, data collection and analysis, decision to publish, or preparation of the manuscript.

==============================
Language WorkBenches (LWBs) are software engineering tools that help domain experts develop solutions to various classes of problems. Some of these tools focus on non-technical users and provide languages to help organize knowledge while other workbenches provide means to create new programming languages. A key advantage of language workbenches is that they support the seamless composition of independently developed languages. This capability is useful when developing programs that can benefit from different levels of abstraction. We reasoned that language workbenches could be useful to develop bioinformatics software solutions. In order to evaluate the potential of language workbenches in bioinformatics, we tested a prominent workbench by developing an alternative to shell scripting. To illustrate what LWBs and Language Composition can bring to bioinformatics, we report on our design and development of NYoSh (Not Your ordinary Shell). NYoSh was implemented as a collection of languages that can be composed to write programs as expressive and concise as shell scripts. This manuscript offers a concrete illustration of the advantages and current minor drawbacks of using the MPS LWB. For instance, we found that we could implement an environment-aware editor for NYoSh that can assist the programmers when developing scripts for specific execution environments. This editor further provides semantic error detection and can be compiled interactively with an automatic build and deployment system. In contrast to shell scripts, NYoSh scripts can be written in a modern development environment, supporting context dependent intentions and can be extended seamlessly by end-users with new abstractions and language constructs. We further illustrate language extension and composition with LWBs by presenting a tight integration of NYoSh scripts with the GobyWeb system. The NYoSh Workbench prototype, which implements a fully featured integrated development environment for NYoSh is distributed at http://nyosh.campagnelab.org.

Introduction

Bioinformatics is a scientific field concerned with storing, organizing and analyzing various types of biological and clinical data and information. Bioinformaticians make daily use of computational abstractions, using a variety of systems, programs and scripting languages to process data. For instance, arithmetic offers abstractions to work with quantities. Similarly, the concepts of strings and sequences are useful abstractions to represent biopolymer molecules. For instance, arithmetic offers abstractions to work with quantities, strings and sequences are useful abstractions to represent biopolymer molecules. Abstractions are similar to ideas or concepts, but in this manuscript, we are concerned with formal abstractions that can be implemented in software. We can find many examples of abstractions in the bioinformatics field. For example, systems built with relational database management systems use the relational data abstraction. Groups who develop ontologies for specific domains rely on specific abstractions for representing knowledge (e.g., OWL (Bechhofer et al., 2004)). Software frameworks extend programming languages with abstractions specific for an application domain (Holland et al., 2008; Stajich et al., 2002). Several bioinformatics frameworks have been developed to improve productivity when writing bioinformatics analysis software (Campagne et al., 2013; McKenna et al., 2010). While frameworks and application programming interfaces (APIs) are useful, they have limitations. Foremost, frameworks and APIs are written in a target programming language and as such, abstractions implemented as a framework must be written in the syntax of this language. This often makes programs more verbose than would be convenient, and requires abstractions to be implemented from scratch for each target programming language. The technology we evaluated in this manuscript overcomes these limitations by making it possible to (i) define new languages specifically tailored to work with their abstractions and (ii) provide mechanisms to transform programs written in the new language into one or more target programming languages.

Language Workbenches are software engineering tools developed in the last ten years that help their users create new languages and the tools to write programs in these languages. Some of these tools focus on non-technical users and provide languages to help organize knowledge. Intentional Software (http://www.intentsoft.com/) is a commercial Language Workbench that falls in this category (Simonyi, 1995; Simonyi, Christerson & Clifford, 2006). Another prominent language workbench aimed at software programmers is the Jetbrains Meta Programming System (MPS) (Dmitriev, 2004; Voelter, 2013; Voelter & Solomatov, 2010). MPS is developed as an open-source project (source code is available at https://github.com/JetBrains/MPS). This workbench was developed to provide the tools necessary to design new programming languages. In our opinion, Language Workbenches (LWBs) represent a true paradigm shift from traditional knowledge management and programming approaches. In this manuscript, we present how we used the MPS LWB to help solve problems related to the development of bioinformatics analysis pipelines. This presentation highlights the advantages of using a modern LWB. The key advantage that we realized in this study is the ability to create very dynamic, interactive program editors that support the script programmers (e.g., bioinformaticians developing analysis scripts) by removing many of the difficulties common with traditional scripting languages. This manuscript provides concrete examples in the results section.

We have recently presented the GobyWeb system to help with analysis of gene expression or DNA methylation data from high-throughput sequencing (Dorff et al., 2013). GobyWeb consists of a web-front end providing access to a compute grid where analyses are performed. The GobyWeb system can be extended to support new types of analysis. Extension consists in developing plugins. Plugins contain a configuration (in XML format) and analysis logic written in the BASH (Bourne Again SHell) language (Fox, 1989; Stallman, 1998). BASH is a widely used scripting language in bioinformatics developed in 1987–1989. The key strength of a scripting language is that it makes it easy to call other programs in order to automate simple analyses, yet is flexible enough than more complicated logic can be expressed if necessary to construct input arguments, process program outputs, or conditionally execute certain steps of the script. While widely used, BASH has limitations that become apparent when trying to develop robust analysis scripts. In this manuscript, we present how we developed a modern alternative to the BASH language with the MPS LWB. We use this example to highlight how LWBs can be used to develop domain or application specific languages that can help bioinformaticians be more productive by using specialized abstractions.

This manuscript is organized as follows. In material and methods, we give a brief introduction to LWB technology and to the MPS LWB. We explain how MPS can be used to create Composable Languages (CLs). In the results section, we then describe the requirements for the design of NYoSh, the Shell scripting language that we use as a working example in this manuscript. We present the CLs that we designed to implement NYoSh in the MPS LWB. We present examples of their use to write GobyWeb plugin scripts. Finally, we discuss the limitations and advantages of LWB technology.

Materials and Methods

Introduction to language workbenches

Readers interested in a general overview of Language Workbenches are referred to the book DSL Engineering (Voelter et al., 2013).

The MPS language workbench

The term “Language Workbench” was coined in 2005 by Martin Fowler (Fowler, 2005) to describe an emerging set of tools to build software using Domain Specific Languages (DSLs). In this project, we used the Jetbrains Meta Programming System (MPS) LWB (Dmitriev, 2004). A deciding factor in choosing MPS among other possible LWBs is that MPS is developed as an open-source project (source code is available at https://github.com/JetBrains/MPS), and offers strong features for creating Composable Languages and the tools needed to work with them. The MPS LWB is distributed with a set of core platform languages. The set of core languages includes BaseLanguage, a language very similar to Java that can be compiled and executed.

Abstract syntax tree

In the traditional programming paradigm, a program is written by a programmer in a Concrete Syntax (CS), specified with a formal grammar, and then translated with a parser into an equivalent memory tree-based representation, called the Abstract Syntax Tree (AST). The AST is a memory representation of a program that can be further manipulated by a compiler or interpreter. The MPS LWB does not use a parser, but instead provides the means to create tools, such as Projectional Editor(s), or PE(s), that make it possible for programmers to directly create and modify the AST in memory. Removing the need for a parser makes it possible to edit programs using several languages in ways that would not be possible with programs written in a concrete syntax.

Language

In the context of the MPS LWB, a language consists of several aspects, including a structure aspect that determines what AST instances can be created in the language, an editor aspect, which makes it possible to specify how the AST should be presented to the programmer, and various other aspects that help implement language behaviors (e.g., actions, constraints, behavior, typesystem, intentions).

Structure aspect

The structure of a language is defined by a set of language concepts. Each concept represents a type of node of the AST. Each concept of a language can have properties (primitive type string, integer, float), children (aggregation) and references (links with 0..1 cardinality) to other concepts. A concept can also extend another concept and implement concept interfaces.

Editor aspect

Working with AST requires that the tree is visualized (projected) in some form to the programmer and she/he is able to modify it. The MPS LWB greatly simplifies the process of developing a robust projectional editor (PE) for a new language. Full language editors are created seamlessly by the MPS LWB. It does so by combining all the mini-editors associated with each type of concept that is defined in a language into an interactive PE.

Generator aspect

MPS supports so called “model to model transformations” and these transformations can be implemented in a generator aspect. Briefly, an AST expressed in language A (a model) can be converted to another AST expressed in language B (the “to model”) via some set of transformations. We developed language generators that translate the concepts of the NYoSh languages into BaseLanguage concepts. In some cases, we developed generators that transform some concept instances into other NYoSh concept instances. For instance, the push and pull GobyWeb language statements are transformed into execute command statements to execute the GobyWeb plugin SDK command line. This was possible because the MPS LWB supports model to model transformations across arbitrary languages.

Language behavior and semantic

The behavior and semantic aspects of a language were implemented with the behavior, typesystem, constraint and intention aspects of the MPS LWB. Semantic error detection was implemented by creating several typesystem “checking rules”.

Environment-aware editor

To create the GobyWeb NYoSh EAE, we developed a dedicated language for modeling environment sources. This language supports the EnvironmentSource and EnvironmentVariable concepts. An environment source is a set of information available at script execution time. By adding a source to the script, the names of environment variables are made available at script design time in the PE. An intention attached to the source makes it possible to reload information from the source. Upon reload, environment variable declarations are created and attached to the AST (as children of the source). Mechanisms for accessing the injected information are provided with the language and misusages and unauthorized accesses to the information are detected and prevented. Auto-completion from the editor assists the programmer to discover which environment variables can be accessed. Sources for importing the local user environment and loading environment definitions from files are also provided with the language. Other languages can define their own configurable environment sources. For instance, the GobyWeb language defines a GobyWeb source that, when added to the script, collects and injects in the editor environment variables extracted from the plugin configuration and the execution platform, allowing to refer to them when writing the plugin script in the PE.

Micro-parsing technique

The results section presents how the micro-parsing technique can be used to work-around some of the difficulties of using a PE to enter BASH commands. The micro-parsing technique consists (1) defining a string property in a concept, (2) extending the concept editor to show the string property (this makes it possible for end users to paste the text into the property) (3) creating an intention that parses the text in the property and modifies the concept or local AST context of the concept according to the content of the text. (4) clearing the string if step 3 did not yield any errors.

Results

Focus on shell scripting

We evaluated the MPS LWB for bioinformatics by focusing on the common problem of developing analysis shell scripts. Writing shell scripts is a popular approach to automate data analyses that need to be performed with several command line programs. We used shell scripts extensively in the GobyWeb system (Dorff et al., 2013). While developing this system, we became aware of some important limitations of BASH shell scripting that hinder programmer productivity when developing new scripts and create serious maintenance problems for existing scripts (other shell scripting dialects than BASH suffer from similar limitations). We describe some of these limitations in the following section.

Limitations of shell scripting

Shell scripts are executed with a Unix Shell interpreter. Unix Shell interpreters like BASH, KSH, CSH, TCSH or Bourne Shell all have common limitations. For instance, shell interpreters:

• Are not compilers – scripting languages are interpreted. This means that if a programmer introduces a syntax or type error in an existing script, it is not possible to detect the error without extensive testing. Indeed, the shell interpreter will only discover and report the error if and when an execution of the script gets to the line that contains the error. This means that when trying to develop robust clinical data analysis pipelines with shell scripts, it is necessary to run the program and exercise all the possible input parameter combinations that could trigger alternate interpretation flows in the script to make sure the script is syntactically valid. Clearly this is non-trivial, time consuming, and sub-optimal when trying to develop robust analysis programs.

• Offer limited tool support – A modern integrated development environment offers features that improve programmer productivity, such as keyword highlighting, syntax error detection, type error detection, code refactoring and intentions. Editors suitable to edit shell scripts at best offer keyword highlighting. Lack of tool support limits the productivity of the bioinformatics script programmer.

• Lack syntax elements for structuring the code – GobyWeb requires that shell scripts defined in its plugins have a structure compliant with an interface (a contract). There is no way to enforce such a constraint with a scripting language.

• Lack mechanisms to promote code reuse – reusable code is code that can be used, without modification, from different programs, or from different parts of the same program. The ability to reuse code is crucial to productivity. Writing (or copying) again and again the same piece of code across multiple programs is not desirable because any problem in the code is duplicated with the code and needs to be fixed in every copy in existence. Object-oriented languages, for instance, provide mechanisms to express contracts and interfaces for components of a program, but these features are lacking from shell languages. Beyond these mechanisms, reusable code needs to be supported also by appropriate tools that, for instance, create standard packages (such as JARs for Java programs) and software repositories (such as Maven) to manage dependencies among packages.

Requirements for an improved scripting language

Considering the limitations of Unix Shell interpreters, we decided to evaluate the MPS LWB by designing an alternative scripting language. To start this process, we assembled a list of requirements that the language should fulfill:

Initially, we created the following list of high-level requirements for NYoSh:

• Composition – in order to test language composition in the MPS LWB, we decided that NYoSh would be implemented by extending existing languages offered by the MPS platform with, small, focused languages, aimed at fulfilling specific programmer needs.

• Abstracting away implementation details – implementation details not affecting the final result of a computation should be hidden from the script programmer as much as possible. We aimed to create reusable abstractions supported by concise, albeit readable notations.

• Ability to execute command pipelines expressed in the BASH syntax. BASH and other shell scripts have very strong language features for executing sets of commands (e.g., pipelines). Since we still use BASH for interactive command development in the shell, we aimed to provide a simple way to import BASH commands into a NYoSh script.

Language design

Figure 1 provides an overview of the tool that we developed to fulfill these requirements. Using the MPS LWB, we were able to develop an integrated development environment (the NYoSh Workbench) that supports scripting by writing programs expressed with composable languages. These languages include:

Figure 1 NYoSh overview and use cases.

Bionformatics pipeline programmers interact with the NYoSh workbench through a projectional editor. The editor allows composing languages and developing solutions. Once a solution is ready, language generators are invoked to produce software to be executed on the target platform. Depending on the kind of solution, the platform is the local computer (for NYoSh scripts) or the compute grid of a GobyWeb system (for GobyWeb plugins). Execution and debugging can be performed within the workbench.

• BaseLanguage – part of the MPS LWB platform, BaseLanguage provides many features found in a general programming language (e.g., variables, loops, object-oriented data structures). BaseLanguage provides most of the capabilities of Java 1.6 (Gosling et al., 2005) and can be seamlessly extended with other languages in the MPS platform such as to offer closures, for instance, which are not offered by the Java programming language.

• New languages that we designed when developing NYoSh:

∘ NYoSh – the scripting language. The NYoSh language has several roles: to provide abstractions to represent command pipelines, to offer named entry points into a script, to offer mechanisms to record script execution and error logs.

∘ GString – a language for working with Groovy-like strings. Groovy strings support lazy evaluation of variable references embedded in the literal. GString adds a similar syntax and functionality to BaseLanguage in the MPS LWB. For instance, writing a GString literal as “${ b}” is possible when a variable b is in the scope of the literal. The GString abstractions make string literals more expressive than the regular BaseLanguage string literals.

∘ PathPatterns – a language that implements pattern matching of file and directory names. Filenames and paths can be matched with wildcards or regular expressions to produce list of filenames.

∘ GobyWeb – an extension of the NYoSh language that provides types of scripts with entry points suitable to write plugin logic for a GobyWeb system.

∘ Environment – a language to manage arbitrary sources of configuration. It allows loading and accessing information from a variety of sources and exposes the loaded information in a uniform way. For instance, environment variables for the process can be loaded and referred to as ${var} where var is the name of the environment variable. The same syntax is used to retrieve the value of a variable available at runtime for a GobyWeb plugin.

∘ TextOutput – a utility language for translating MPS concepts to text output. This language is used when generating configuration files for GobyWeb plugins.

Each of these languages provides a set of abstractions designed for a specific purpose. This is an advantage because it helps keep each language small and focused. When combined through language composition, these focused languages provide complementary features and make it possible to write NYoSh programs as concise as BASH scripts, albeit compiled and with strong development environment support. Figure 2 shows a snapshot of the NYoSh workbench, an integrated development environment that supports the development of NYoSh programs. The languages presented in Fig. 1 are bundled in this environment.

Figure 2 The NYoSh workbench.

The workbench is a graphical Integrated Development Environment allowing to fully manage the development lifecycle of solutions based on the included languages. On the left side, a navigation panel allows to browse the Abstract Syntax Tree (AST) of the composed language concepts. On the right side, the editor shows the text-like projection for a specific script (this script is the rendering of the AST shown in the left panel, but this is transparent to the workbench programmer). At the bottom, the version control console reports messages related to the source control operations performed. Also shown is a Git commit dialog to illustrate that the workbench provides full source control integration (with Git, or SVN). The NYoSh Workbench desktop application shown in this figure was generated by the MPS language workbench and packaged for distribution across multiple operating systems (MacOS, UNIX/LINUX, Windows).

In the next sections, we describe a few of these languages in detail to illustrate some of the advantages of LWB technology.

Language composition

Figure 3 presents an example of language composition. In this figure, we present a language concept diagram to illustrate the relationships among concepts of different languages. The diagram follows the convention of the Unified Modeling Language, a popular graphical modeling language (Rumbaugh, Jacobson & Booch, 1999). Each grey or blue box in Fig. 3 groups the concepts that belong to a language (the color blue highlights core languages provided with the MPS platform). See material and methods for a brief description of MPS languages and concepts. Inheritance (or generalization) relationships across concepts of two languages are a primary mechanism to compose these languages. For instance, Fig. 3 shows that the GString concept of the GString language extends the Expression concept of BaseLanguage. This makes it possible to use instances of the GString concept everywhere that an instance of the BaseLanguage Expression concept is expected. Reference is another mechanism to compose languages through reuse. In Fig. 4, we present the same language composition from the point of view of the script programmer. In this example, we show how a GString instance can be used as initializer to a variable declaration of type string. This is possible because of the is-a relationship between the GString concept and the fact that variable initializers are of type Expression in BaseLanguage. This example of language composition would not be possible with traditional programming language technology and is a key motivation for using a LWB. It is important to note that the script programmer can determine which languages can be used in the projectional editor on a program-by-program basis.

Figure 3 Composing the GString language with other languages.

This diagram presents GString language composition form the point of view of the language designer. The language the concepts belong to is shown in a grey or blue box labeled with the name of the language. The color blue is used for language(s) provided by the MPS platform. In the GString language, the GString concept extends the Expression concept from (the) BaseLanguage (language). This extension relationship makes it possible to use instances of the GString concept wherever Expression instances are expected. A GString concept contains GStringComponent children (aggregation link). GStringComponent is an abstract concept whose concrete sub-concepts, GStringLiteral and GStringReference specialize (generalization link) the component. GStringLiteral represents a plain string. GStringVarReference represents a reference to a variable inside a GString. GStringVarReference references a VariableDeclaration from BaseLanguage (reference link). Specializations can also occur across languages, as shown with the VariableReaderGStringComponent specialization in the Environment language. This specialization makes it possible to use environment variables inside a GString (see Fig. 4 for an example of how these languages are used from the point of view of the script programmer).

Figure 4 GString composition from the point of view of the script programmer.

Assembling a string instance to display a message can be done in pure BaseLanguage syntax as shown in panel (A). This code fragment concatenates string literals with the + concatenation operator. An environment variable value is retrieved for the variable USER and inserted in the string. The code is similar to the Java syntax and a bit verbose. (B) In the second panel, we show the same string constructed with BaseLanguage and the GString language extension. The syntax is more concise than in panel (A), yet will generate equivalent code. The (B) snapshot from the editor shows how GString instances appear to the script programmer in a textual representation that many programmers will find natural. (Concepts are underlined with color in this figure to clearly indicate which parts of the program belong to which languages. Color underlining does not appear in the NYoSh editor, but text color is as shown). In this example, a GString instance is assigned to a string variable called composedString. This is possible because the GString concept extends the Expression concept of BaseLanguage (see Fig. 3). The GString instance shown to the right of the equal sign (=) contains four GStringComponent instances. The first one is a GStringLiteral (This is the), the second a GStringVarReference to variable in scope (${name}), the third another GStringLiteral (You are logged in as), and the fourth a VariableReaderGStringComponent (${USER}) that references an environment variable. This figure illustrates that LWB make it possible to create new syntax constructs that work seamlessly with other languages.

Interactive editing

The GString language, like other languages developed with a LWB, offers more than just a concise syntax. For instance, we were able to endow the language with Intentions. An intention is simple method to extend the user interface of the Projectional Editor (PE, see Material and Methods) by adding menu items that are shown to the end user when the cursor is positioned in the editor on certain instances of a concept. Selecting the menu item of the intention applies some changes to the program(s) under editing. Changes can be made to the AST being displayed in the window, but can also create other ASTs if necessary (e.g., the equivalent or creating new files in a text editor). Intentions can be defined for the concepts of a language, or for concepts of other languages. Another substantial advantage of LWB and PE is that it is possible and straightforward to add semantic error detection to a language. This capability is absent from traditional languages and can only be added at great cost by creating an integrated development environment for each language. In a LWB, detecting errors in a language is as simple as writing logic to detect error conditions in the AST. The PE takes care of highlighting the part of the program presentation and displaying the error message to the programmer. The MPS LWB makes it possible to build an interactive PE in a very modular manner.

Creating a command execution statement

A LWB makes it possible to seamlessly extend a language with new language constructs. We took advantage of this capability by developing an ExecuteCommand concept in the NYoSh language. We designed the ExecuteCommand concept to provide a LWB equivalent to BASH command pipelines. Figure 5 presents the concept diagram for this language extension. ExecuteCommand extends the BaseLanguage Statement concept, and can therefore be used everywhere in programs where a statement would be expected. Figure 6 shows how the execute command appears to the script programmer. In designing the syntax of the ExecuteCommand, we aimed for a similar expressiveness than that experienced by BASH script programmers. Commands can be separated by the familiar command binary operators (i.e., | & ‖ && ;) and these operators provide the same semantic as the BASH interpreter at runtime. Semantic errors (such as two consecutive binary operators) are recognized and highlighted in the PE.

Figure 5 ExecuteCommand: composition with other languages.

The ExecuteCommand concept is designed to be a container (aggregation link) of AbstractCommand instances. A sub-concept of AbstractCommand defines a concrete command to be executed. NYoSh provides three basic extensions: (1) GStringCommand, allowing to directly type commands with GString instances, (2) BynaryOperator, an abstract concept for defining BASH command separators, (3) ConsumeOutput, another abstract concept to be extended for redirecting the output of the preceding commands (for instance in a file, as showed in the figure). Other languages can provide their own commands. For instance, the GobyWeb language defines two AbstractCommand extensions: FetchCommand (to obtain the names of input files for a plugin) and PushCommand (to push output files produced by a plugin to the GobyWeb system). These commands are offered as Commands to make it possible to use them in command pipelines.

Figure 6 ExecuteCommand from the point of view the script programmer.

A fragment of NYoSh program is shown with two statements. The first statement is a variable declaration statement and the second an execute statement. Note that an execute statement contains a list of commands separated by binary operators. In this example, three GStringCommand concept instances are shown, separated by a pipe operator. The last command of the list is an instance of RedirectToFile concept configured to write to an md-alignment.sam filename. GStringCommand benefits from the ability of GString to access environment variable values to construct each command.

Language compilation

We have implemented language generators that make it possible to compile NYoSh programs into pure Java programs. While MPS BaseLanguage is already endowed with a suitable generator, any language extension introduced in a language must reduce the AST to the target implementation language with a custom generator. When a programmer (re)builds a NYoSh program, the NYoSh Workbench invokes all the language generators and produces target platform files. With the set of generators that we developed, NYoSh programs are compiled to one or more Java classes or Java Archive (JAR). All extended language constructs are reduced to the Java 1.6 language and to a small set of open-source Java libraries. The GobyWeb language includes generators that also create the shell scripts necessary to execute the NYoSh JAR files in a GobyWeb system and build system rules to deploy these files automatically to a GobyWeb plugin repository.

Micro-parsing technique

In working with the MPS LWB, we devised a technique that can sometimes facilitate the development of programs in the projectional editor. We used this technique successfully for two types of data entry. Figure 7 illustrates the micro-parsing technique when entering BASH commands with references to variables (A–D) and when entering BASH commands that contain operators (E–F). The technique is useful to enter code or semi-structured text fragments that would be tedious to enter directly with the projectional editor. Briefly, the micro-parsing technique consists in storing text in a string property of the concept and to create an intention that parses the text, adjusts the AST according to the content of the string, and clears the string. In Fig. 7, the text of a BASH command line is entered in the text property of a GStringLiteral. The intention parses the text to extract a sequence of string literals and variable references. It then replaces the GStringLiteral with an equivalent list of GStringComponent sub-concepts. Finally, the intention also defines variables for each GStringVarReference immediately before the statement that contains the GString. In our experience, the micro-parsing technique speeds up entry of existing BASH commands into a NYoSh script. We have also used the technique in the TextOuput language to capture mostly unstructured text in a property, parse it at new lines, and parse the text of select lines at a character delimiter. Since the intention is applied on select lines only, only these lines need to be parsed into Phrase concept instances. Replacing text phrases with Phrase concept instances makes it possible to create text output that varies with the properties of an input concept (this is done in MPS by creating a mapping rule and template in a generator aspect). This technique is simpler to implement than parsing every word of the unstructured text and representing it as a concept instance.

Figure 7 The micro-parsing technique helps enter BASH commands into NYoSh.

This figure illustrates how intentions can be used to facilitate inputting or reusing complicated BASH command lines into a well-formed NYoSh program. Consider the BASH command line shown in (A). This command line would typically have been developed interactively in the BASH interpreter, possibly trying the command with test data until it functioned as desired. (B) Here we pasted the text of the command as a GStringLiteral and assigned it to a variable (bashCommand). (C) Activating a pre-defined GString intention (Extract ${var} as variables) converts the literal into (D) a GString where variable references in the format ${name} have been replaced with references to program variable declarations. The intention implements a micro-parsing operation that substantially simplifies the conversion of BASH commands into NYoSh scripts. As expected, changing the value of the variables before the GString evaluation will change the value of the literal produced. A similar micro-parsing technique is illustrated in panel (E) where we pasted a BASH command pipeline into a NYoSh execute statement. Activating the micro-parsing intention called “Parse literal into command expressions” parses the sequence of individual commands and operators and generates a corresponding AST fragment in the execute statement. If each fragment contained a ${var} pattern, the intention shown in (C) could be used to introduce the corresponding variables before the execute statement. The micro-parsing technique greatly simplifies adapting existing BASH command pipelines to NYoSh.

NYoSh and GobyWeb: integration through composition

Figure 8 illustrates how we were able to combine NYoSh scripts with the GobyWeb system. The NYoSh script logic for a GobyWeb aligner plugin is shown. We achieved a tight integration through language composition. The MPS LWB supports embedding the editor of one concept in the editor of another concept in a straightforward manner. We took advantage of this capability to provide a structured header on top of a GobyWeb NYoSh script that makes it possible to enter information about where the script belongs in the GobyWeb plugin system. Users of the NYoSh workbench can create new GobyWeb NYoSh scripts by selecting the type of plugin among the GobyWeb supported plugin types (aligner, alignment analysis, resource, artifact installation script, task). A GobyWeb NYoSh script template is then created that provides the empty entry points needed to implement the type of plugin specified. This ability to provide pre-configured scripts is a strong substitute for the usual practice of providing text templates for empty scripts and another advantage of using a LWB. Figure 8 also illustrates how the different languages we have developed are composed with BaseLanguage and Java libraries to write a complete script. Rebuilding this script in the NYoSh workbench automatically compiles and deploys the jar packages to the GobyWeb plugin location specified in the header. We have provided in supplementary material the complete listing of the configuration files and Java source code that are generated when the script shown in Fig. 8 is compiled in the NYoSh Workbench. Comparing the script in Fig. 8 and the generated Java source code in supplementary material provides a compelling justification for the design of concise domain specific languages: the NYoSh script can be much more concise because the notations of its languages encapsulate many of the details that would need to be explicitly stated in a program written in the Java language. We note that the Java program uses several libraries that we developed to support the runtime behavior of a NYoSh script. For instance, these libraries provide ways to run pipelines of commands, or to interact with the GobyWeb runtime environment. The syntax of the NYoSh languages hides the complexity of calling these runtime libraries from the script programmer.

Figure 8 Tight integration between NYoSh scripting and the GobyWeb plugin system achieved with language composition.

Using the MPS LWB, we have been able to tightly integrate NYoSh scripts with the GobyWeb plugin system. Specifically, users of the NYoSh workbench can use NYoSh scripts to implement GobyWeb plugins logic (1) At the top of the editor, the GobyWeb plugin information can be provided to associate the script (dashed box) to a specific GobyWeb plugin. The fields are defined by the GobyWeb language structure and just need to be filled in via the editor. Once the programmer has filled in the information, the NYoSh script becomes aware of the environment the plugin will execute in. This plugin script illustrates the tight integration that can be achieved between languages in the MPS LWB. The GobyWeb language extends the NYoSh language and adds new structure not present in NYoSh scripts. (2) For instance, the error management information is specific to GobyWeb (it can be added with an intention). (3) GobyWeb script entry points are created with instructions that load the plugin environment into the script. The GobyWeb source loads the plugin environment using information provided in (1). (4) Java libraries can be used in the script (shown is using the Apache Commons-IO open source library). (5) GString is used to create string values that embed values from the plugin runtime environment. (6) GStrings values can be used as environment variable names in a concise syntax. (7) NYoSh execute statements provide a convenient way to execute programs in the script. The arrow indicates that execute statements use the current environment, which was modified by the load environment sources statement.

Summary of LWB drawbacks

We found a few drawbacks when working with the MPS LWB. The first and most noticeable drawback is that LWBs require the programmers to adapt and learn how to work with a projectional editor (PE, see Material and Methods). In our experience, the process can be particularly frustrating for about half a day, in this period of time when experienced programmers will wonder why they are putting themselves through the pain of learning how to enter code in the PE when they were perfectly efficient with a text editor. As experienced programmers, we started to feel quite comfortable with the PE of the MPS LWB in about one week. The second important drawback of LWB technology is that programs are persisted in data structures that cannot be easily edited without tool support. This makes the stability of the specific LWB tool of primary importance when programming with a LWB, because if the LWB fails to persist correct data structures to disk, the programs may become corrupted and non-editable. We only encountered minor problems of this kind on two occasions with the MPS LWB, presumably because we were working with an early access program release (EAP 3.0). In each case, we were either able to fix the problem by (1) editing the model files directly, or (2) recovering an earlier version of the program in source control repository. The last drawback is specific of the MPS LWB. We noted the absence of dependency management capability in the MPS tool. Integration with tools such as Maven or Ivy would be useful to develop projects that reuse many components, Java packages and languages, but is currently lacking from the MPS LWB.

Summary of LWB advantages

Figure 9 provides a summary of important and unique advantages of LWB technology. These advantages include: use of intentions, seamless execution and debugging, domain and application-specific error detection and messages, and code completion.

Figure 9 Key Advantages of the NYoSh Workbench.

Because the NYoSh Workbench was implemented with the MPS LWB, it benefits from the capabilities offered by this robust LWB (panels A–D). (A) The workbench supports intentions, which are interactive ways to modify the AST being edited. In NYoSh, for instance, intentions are used to automate the creation of boiler-plate code, as shown here, or to implement the micro-parsing technique shown in Fig. 7 (B) NYoSh scripts can be executed or debugged seamlessly from within the workbench. This capability makes NYoSh scripts behave almost as if they were interpreted because although compilation is required, it is triggered automatically when the program has changed and the programmer requests execution. (C) We implemented domain and application specific error detection to provide syntax error highlighting. In the example shown, the error message indicates that the GobyWeb environment requires a valid plugin configuration (location must be filled in under “plugin system”). (D) At any point in a NYoSh script, the programmer can activate auto-completion in the editor (control+space key combination). The auto-completion dialog then suggests which concepts are valid in the specific context being edited. In the example shown, auto-completion is invoked after entering an execute statement. The completions offered include available sub-concepts of AbstractCommand commands in the languages imported in the editor. (E) In NYoSh, we have used auto-completion and intentions to implement an environment-aware editor (EAE). An EAE is an editor that is aware of the environment in which the program will execute, and offers completion suggestions to the programmer that are adapted to the target environment. In panel (E, top and bottom), we show how the NYoSh EAE is aware of the environment variables available to a GobyWeb plugin when the plugin will execute; (E, middle) auto-completion suggests the output slot names that are defined in the GobyWeb plugin that the programmer is working on (slots represent the possible outputs of a plugin).

Environment-aware editor

Using these abilities we were able to create an environment-aware editor (EAE) for GobyWeb NYoSh scripts. Figure 9E shows that the GobyWeb NYoSh EAE makes it possible to:

- View environment variable names at the time when the script is being designed (design time). Variable names are contributed both from the JVM environment and by the GobyWebSource concept, which is aware of resources available to the specific plugin being developed (GobyWeb resources are a special kind of plugin that encapsulates software or data installed on the compute nodes (Dorff et al., 2013)).

- View the name of input/output slots in the editor that are defined in the GobyWeb plugin configuration file.

Discussion

Domain and application specific languages

Languages have been designed to facilitate the development of specific application programs or for programming in domains where programs share similar requirements. In the high-performance scientific domain, the Swift Parallel Scripting language has been developed to facilitate the parallelization of large-scale data processing on compute grids (Wilde et al., 2011). Swift provides a compelling motivation for developing domain-specific languages when the language can abstract the complexity of parallel processing. Swift was developed with traditional programming language technology and therefore does not benefit from an integrated development environment or any of the LWB advantages we highlighted in this manuscript. In the bioinformatics domain, AndurilScript was developed as part of the Anduril project to provide a concise language to express the logic of data processing components (AndurilScript is an application-specific language) (Ovaska et al., 2010). Both Swift and AndurilScript illustrate how domain or application specific scripting languages can increase the productivity of programmers working in a particular domain or with a specific application program. Because Swift or AndurilScript were developed with traditional programming language technology it would be cumbersome to reuse parts of the implementation of these languages to develop languages with related requirements (such as extending Swift with a new language statement type). This is not the case with languages developed with LWB technology since language extension and composition is the raison d’être of these tools.

Who needs another programming language?

A large number of programming languages have been developed along the years and it is reasonable to ask if there is a real need for more languages. The last sixty years since the design of FORTRAN have seen a proliferation of programming languages that mostly provide similar computing abstractions, while innovating in one or a few aspects, and that are difficult to use together (e.g., using a combination of Python, Perl and Java to develop an analysis program is neither practically simple, nor very useful, because the languages mostly share similar capabilities). LWBs make it possible to develop new languages, but their most important characteristic is that they seamlessly support the design and use of new languages. This is a key point because new languages can be small and focused on providing a small number of unique abstractions. Focus is possible since features missing from a small language can be obtained by composing this language with other languages that provide complementary features. The composition mechanisms offered by LWB greatly facilitate language reuse and separation of concern between languages. We have provided examples of this throughout this manuscript. For instance, we did not create completely new scripting language syntax, but instead reused the capabilities of the robust BaseLanguage implementation provided by the MPS system. We extended BaseLanguage with small languages that provide the abstractions that were needed to produce the unique functionality that our application required. We expect that widespread use of LWB will result in a multiplication of small languages, but in a manner that will increase language reuse and interoperability, rather than in the historical language fragmentation that has been observed with traditional language technology.

Increased productivity

Most Unix Shell interpreters have been developed over many years (for instance, BASH was developed across a period of two years from 1987 to 1989). In 2013, using LWB technology and the MPS platform, we were able to assemble a proof of principle of an alternative scripting language in about two months and fine tune this prototype in another two months (see timeline in Supplementary material File, Fig. S1; two people worked on the project at about 50% effort for about four months). Since our team of two was new to the MPS platform at the beginning of the project, it seems clear that the MPS LWB provided increased productivity for this project. A clear factor in increasing our productivity was that the MPS platform offers strong mechanisms for reusing and composing languages. These mechanisms, illustrated in the results section of this manuscript, made it possible to extend a fully featured general programming language already offered by MPS (BaseLanguage) with capabilities that make the language feel more like a scripting language. The concise syntax of the NYoSh languages helps programmers increase productivity when reading and writing scripts.

Differences with traditional languages

Python and Perl are programming languages also frequently used in bioinformatics to automate analyses. Importantly, Python or Perl programs can be compiled, thus helping to avoid many of the limitations of shell script interpreters. However, similarly to other traditional programming languages, neither Python nor Perl can be extended with new language syntax to offer domain dependent abstractions of the type that we were able to create with LWB technology. It is also usually not straightforward to endow text editors with semantic error detection for these languages.

Software frameworks

Software frameworks can be developed to extend general programming languages with abstractions such as data structures and algorithms adapted to a scientific field or application domain. Languages developed in a LWB support a similar extension, but have a key advantage: they can provide new language constructs that integrate seamlessly with the rest of the language. This is not possible with traditional programming languages where frameworks must be expressed using the syntax of the host language.

Projectional editor

A projectional editor is different from a textual editor and most users need a few weeks of practice to become familiar with the editor. While we have found that the process of adapting to a PE can be frustrating, we have found that after a period of intense development with the MPS PE, switching back to a text editor can also be frustrating. Overall, we felt that the longer-term advantages of using LWB technology far outweigh the minor discomfort and effort needed when switching from a text editor to a PE. Based on anecdotal evidence, we would hypothesize that users who have never worked with a text editor for programming tasks may find it easier to learn a PE that programmers who have only worked with text editors for a long time. We believe this would be possible because a PE offers a more controlled environment where mistakes are harder to make and also offers mechanisms that greatly enhance interactive development (such as intentions and semantic error detection). In a sense, the projectional editors are closer to the user interfaces that non-technical users are already familiar with than they are to the text editors that more experienced programmers are used to. Testing this hypothesis would require a well-designed human subject study and is beyond the scope of this manuscript.

More human readable programs

When designing language constructs for NYoSh we found that PE technology makes it possible to present language constructs as English phrases. For instance, the ExecuteCommand statement is rendered as “execute: <pipeline>”, the EnvironmentSource statement is rendered as “load environment sources”, the GobyWeb error management attribute is called “error management:”. We think that these presentation choices make programs more readable by humans. Traditional programming languages have been designed around the constraints of Lexer and Parser technology, which is not compatible with the use of natural language-like phrases. LWBs do not have these limitations because they work directly with the AST (see Materials and Methods). We recommend using natural-language-like phrases to design languages aimed at less technical users.

Conclusion

By and large, we found that the advantages of LWB technology outweigh their drawbacks. Using this technology, we were able to quickly build a set of composable languages that together provide a replacement for shell scripting. We were able to take this language one step further and integrate the new languages tightly with GobyWeb. In this process, we created an environment-aware editor that facilitates the writing of GobyWeb plugin script logic (because the editor can provide automatic completion for aspects of the development that otherwise would require frequent lookups in the documentation). Through this experience, we feel as if we have looked into the future of programming, and saw a technology that is ripe for widespread adoption. We think that because LWBs make it possible to design new languages with their own syntax, they are a great fit for scientific programming problems where programmers frequently need to devise and compute with new abstractions. This is particularly true in the field of bioinformatics where many types of abstractions are developed to compute about biology.

Supplemental Information

Figure S1 Project timeline and generated source code

Click here for additional data file.

Additional Information and Declarations

Competing Interests

Author Contributions

The authors declare no competing interests.

Manuele Simi performed the experiments, analyzed the data, wrote the paper.

Fabien Campagne conceived and designed the experiments, performed the experiments, analyzed the data, wrote the paper.

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
