# Peer review of "Composable languages for bioinformatics: the NYoSh experiment"

_PeerJ, doi:10.7717/peerj.241_

## Round 0.1 · original submission · Minor Revisions

· Academic Editor

Minor Revisions

Both reviewers found your paper compelling and of general interest. Both have also provided a number of comments that I believe will help improve the readability of your paper. Please consider them carefully as you revise your paper.

·

Basic reporting

-line 52, the argument in the sentence "This often makes programs more verbose than would be convenient." is, in my opinion, less an issue that implementing the same solution in multiple languages, which LWBs solve nicely.

-I highly encourage the authors to include a timeline or table describing how long it took them to develop each piece of NYoSh (composable languages, language compilation modules, etc.). This is valuable information that other bioinformaticians would benefit from as they weight the benefits and drawbacks of using NYoSh and developing their own solutions in a LWB.

Experimental design

No comments

Validity of the findings

-lines 369-371, the justification referred to in this sentence needs to be unpacked and/or qualified: "Comparing the script in Figure 8 and the pure-Java source code equivalent in supplementary material provides a compelling justification for the design of concise domain specific languages." Clearly the script is going to be shorter—and likely easier to read—because it's using the Java libraries, while the Java implementation is not using any additional libraries and must implement/define everything (and is doing so in an automated way that may not optimize readability). The authors should be explicit about the advantages of the script vs. Java implementation.

Additional comments

-Sentence on line 38-39 does not make sense.

-It's very unusual for Figure 7 to be the first figure mentioned (line 161); this could be fixed by omitted reference to Figure 7 until later.

Reviewer 2 ·

Basic reporting

No Comments.

Experimental design

No Comments.

Validity of the findings

No Comments.

Additional comments

* "that support the script programmers"
-> it is not clear to me who exactly the target audience is.
what kinds of scripts are these?
* "analysis logic written in the BASH language
-> that is not the shell script bash thing, right?
* Usually, Language workbenches are abbreviated LWB.
* "creating several typesystem “checking rule”
-> 's' missing in "rules" at the end.
* "Micro-parsing Technique.
-> it is not clear why this is important and how it fits in here.
* Reading the Results section, it seems like BASH really is the
shell thing. Did that really originate from bioinformatics as you
say earlier? I didn't know that.
* "robust clinical data analysis pipelines with shell scripts
-> it is somehow a bit sad that shell scripts is the state of the art
for this kind of data analysis. I can clearly see why you would
want to develop something more advanced/robust.
* "reusable code is code that can be used, without modification, from
different programs
-> or from diff. parts in the same program.
* "Beyond the structure, reusable code needs ...
-> even more importantly, it has to have clear interfaces and contracts.
* Is there a reason why all figures are at the end? THat's a bit inconvenient
during reading.
* "BaseLanguage provides most of the capabilities of Java 1.6
-> in fact, it has more than Java (closures, e.g.)
* "a specialization of the NYoSh language
-> I think you hadn't use "specialization" before, you used extension.
Difference?
* "See material and methods for a brief description of MPS languages
and concepts
-> should this be a footnote? Interrupts reading flow.
* " the same semantic than BASH at..."
-> "the same semantics as a BASH at..."
* The micro-parsing stuff is interesting. You could automate it further, by
having checking rules that automatically detect a mismatch between
te extracted
AST and the textual string.
* The comparison between micro parsing and the multiline/richtext stuff isn't
quite to the point, because the important thing about the rich text stuff is
the ability to embed actual MPS program notes into free-flowing text.
* "developed along the years
-> "developed over the years

---

## Round 0.2 · accepted · Accept

· Academic Editor

Accept

Thank you for your careful revision and incorporation of the helpful comments from reviewers.